# Efficacy of Zinc Fortified and Fermented Wheat Flour (EZAFFAW): A randomized controlled trial protocol

Jai K. Das[1,2]*, Zahra Ali Padhani[1], Muhammad Khan[1], Mushtaq Mirani[1], Arjumand Rizvi[3], Imran Ahmed Chauhadry[3], Rahima Yasin[1], Tariq Ismail[4], Saeed Akhtar[4], Kehkashan Begum[5], Junaid Iqbal[5], Khadija Humayun[6], Hamna Amir Naseem[1], Kauser Abdulla Malik[7,8], Zulfiqar A. Bhutta[1,9]

1 Institute for Global Health and Development, The Aga Khan University, Karachi, Pakistan, 2 Division of Women and Child Health, The Aga Khan University, Karachi, Pakistan, 3 Centre of Excellence in Women and Child Health, The Aga Khan University, Karachi, Pakistan, 4 Institute of Food Science and Nutrition, Bahauddin Zakariya University, Multan, Pakistan, 5 Nutrition Research Laboratory, The Aga Khan University, Karachi, Pakistan, 6 Department of Paediatrics and Child Health, The Aga Khan University, Karachi, Pakistan, 7 School of Life Sciences, Forman Christian College (A Chartered University), Lahore, Pakistan, 8 Pakistan Academy of Sciences, Islamabad, Pakistan, 9 Center for Global Child Health, Hospital for Sick Children, Toronto, Canada

* jai.das@aku.edu

**Data Availability Statement:** This is a study protocol, and no data is generated. All the data will

## Abstract

### Background

Zinc deficiency poses significant health risks, particularly in low-income settings. This study aims to evaluate the impact of agronomically zinc biofortified (fermented and non-fermented) and post-harvest wheat flour flatbread on zinc status and metabolic health in adolescents and adult women in rural Pakistan.

### Methods

A four-arm triple-blind randomized controlled trial will be conducted in a rural district of Pakistan. Participants (adolescents aged 10–19 and adult women aged 20–40) will be assigned to receive fermented or unfermented high zinc agronomically biofortified wheat flour flatbread, post-harvest zinc-fortified wheat flour flatbread, or low zinc conventional whole wheat flour flatbread. The meal would be served once a day, six days a week for six months. The study aims to enroll 1000 participants and will be analyzed based on the intention-to-treat principle. The trial is registered with number NCT06092515.

### Outcomes

Primary outcomes will include serum zinc concentration and metabolic markers, while secondary outcomes include anthropometric measurements, blood pressure, and dietary intake.

be shared in the final manuscript and supplementary files.

**Funding:** This trial is funded by the "International Food Policy Research Institute (IFPRI) - (HarvestPlus Program)" and the grant number is "2022H8416.AKU". The funders had no role in study design, data collection and analysis, decision to publish, or preparation of the manuscript.

**Competing interests:** The authors have declared that no competing interests.

## Conclusion

This trial will provide valuable insights into the efficacy of agronomically zinc biofortified wheat flour in improving zinc status and metabolic health. Findings may inform public health strategies to combat zinc deficiency in resource-limited settings.

## Introduction

Zinc is integral to numerous essential metabolic pathways and usually manifests as a rather nonspecific deficiency, with varying severity and varies by age [1]. In childhood, zinc deficiency appears as growth retardation and cognitive impairment [2, 3], recurrent infections including diarrhea [4], loss of hair, conjunctival and eyelid inflammation. In adolescents and adults, zinc deficiency can lead to fertility issues, reduced work capacity and metabolic disorders [5, 6] whereas for the elderly population, recurrent infections are a common manifestation [7].

Zinc supplementation, administered in pharmacological doses via pills or syrups, has demonstrated several positive effects in different age groups. In children, zinc supplementation suggests improvement in mean serum zinc concentration [8, 9] and reduction in the incidence of diarrhea and its morbidity but has shown no effect on pneumonia and malaria morbidity [10–12] and evidence also suggests a small improvement in height [8, 9]. There is no convincing evidence for the effectiveness of zinc supplementation on pregnant women except for a small effect on preterm births in low- and middle-income countries [13]. In adults with insulin resistance, zinc supplementation does not appear to prevent the onset of type 2 diabetes [14], while zinc supplementation for individuals aged 55–87 has shown to reduce the incidence of infections and improve plasma zinc concentration [15]. The potential impact of zinc on atherosclerotic disease remains inconclusive, despite its theoretical benefits in reducing plasma lipid peroxidation end products and endothelial cell adhesion molecules [16]. A recent meta-analysis suggests that low-dose, long-term zinc intake from supplements, and potentially biofortification, may offer benefits in terms of risk factors for type 2 diabetes (T2DM) and cardiovascular disease (CVD) [17].

Zinc bioavailability from a mixed or vegetarian diet based on refined cereal grains is estimated to be 26–34%, whereas 18–26% is absorbed from an unrefined cereal-based diet [18]. The actual amount of absorbed zinc not only depends on the zinc content of the consumed diet but is highly affected by its intestinal zinc bio-accessibility and bio-availability [18]. The inhibitory effect of phytate on zinc absorption is concentration-dependent and has shown to be more important than the phytate content of the product itself [19]. Fermentation theoretically results in higher zinc bioavailability if a significant reduction of phytate is achieved.

Food fortification with zinc appears as an attractive public health strategy, with numerous programs initiated in developing countries as a cost-effective way to combat zinc deficiency [1, 12]. Systematic reviews evaluating zinc fortification indicate significant improvements in plasma zinc concentration, reduced prevalence of zinc deficiency, increased child weight, and enhanced short-term auditory memory. However, data for adolescents, pregnant women, and lactating women are limited [20]. Biofortified zinc flour have also shown mixed effects [21].

The aim of this randomized controlled trial (RCT) is to assess the effect of fermented and unfermented wheat flour flatbread from agronomically zinc biofortified and post-harvest fortified wheat flour when compared to conventional low-zinc wheat flour flatbread and its impact on health including zinc status, anthropometric outcomes, risk of T2DM and morbidity on

adolescent and adult females. In Pakistan, common naan is a type of flatbread, is typically made from fermented maida (all-purpose low extraction) wheat flour, characterized by refined flour with minimal bran and germ. However, in most households, the community usually uses unfermented whole wheat flour flatbread. Wheat samples from various sites across Pakistan were tested, with zinc levels ranging from 20-50mg/kg. The trial chose two wheat varieties: conventional low zinc (20-25mg/kg) and high zinc content which is agronomically biofortified (>35mg/kg) which was grown under collaborator supervision at the study site.

## Materials and methods

### Objectives

The objective of this trial is to assess the effect of whole wheat flatbread made from agronomically biofortified 'high zinc wheat'—(fermented and unfermented) and 'post-harvest zinc-fortified wheat flour' compared to conventional 'low zinc' whole wheat flatbread on zinc status and metabolic health in adolescents (10–19 years) and adult women (20–40 years).

### Study design

This would be a four arm individually randomized, triple-blind trial conducted in a rural district of Pakistan. The study will adhere to the guidelines outlined in the Consolidated Standards of Reporting Trials (CONSORT) for RCTs [22]. The trial is registered at clinicaltrials. gov with number NCT06092515 [23]. Fig 1 highlights the SPIRIT schedule of the trial (S1 Checklist).

### Intervention

Participants will be randomly divided into four groups according to the four-arm design in a 1:1:1:1 ratio (See **Fig 2** for study groups).

- Group 1: will receive fermented high zinc agronomically biofortified wheat (>35mg/kg) flour flatbread.

| | | | STUDY PERIOD | | | | | | | | |
|---|---|---|---|---|---|---|---|---|---|---|---|
| | Enrollment | Allocation | POST ALLOCATION (INTERVENTION) | | | | | | | | CLOSE-OUT |
| **Timepoint** | -T1 Baseline | 0 | T1 Month-1 | T2 Month-2 | T3 Month-3 | T4 Month-4 | T5 Month-5 | T6 Month-6 | Endline | |
| **ENROLLMENT** | | | | | | | | | | | |
| Base line | X | | | | | | | | | | |
| HH survey/Dietary data | X | | | | | | | | | | |
| Anthropometric Assessment | X | | | | | | | | | | |
| Biochemical Screening | X | | | | | | | | | | |
| Sample Calculation | X | | | | | | | | | | |
| Randomization | X | | | | | | | | | | |
| Informed Consent | | X | | | | | | | | | |
| Enrollment | | X | | | | | | | | | |
| Allocation in groups | | X | | | | | | | | | |
| **INTERVENTION** | | | | | | | | | | | |
| Setting up study kitchen | | X | | | | | | | | | |
| Food delivery to participants | | | ←——————————————————→ | | | | | | | | |
| Daily food intake | | | ←——————————————————→ | | | | | | | | |
| Monitoring BP | | | ←——————————————————→ | | | | | | | | |
| Monitoring Morbidity | | | ←——————————————————→ | | | | | | | | |
| **MIDLINE** | | | | | | | | | | | |
| Dietary data | | | | | X | X | | | | | |
| Biochemical screening | | | | | X | X | X | | | | |
| **ENDLINE** | | | | | | | | | | | |
| HH survey/Dietary data | | | | | | | | | X | | |
| Anthropometric assessment | | | | | | | | | X | | |
| Biochemical Screening | | | | | | | | | X | | |
| **POST INTERVENTION** | | | | | | | | | | | |
| Date Analysis | | | | | | | | | | | X |
| Data Management | | | | | | | | | | | X |
| Publications | | | | | | | | | | | X |

**Fig 1. SPIRIT schedule of EZAFFAW trial.**

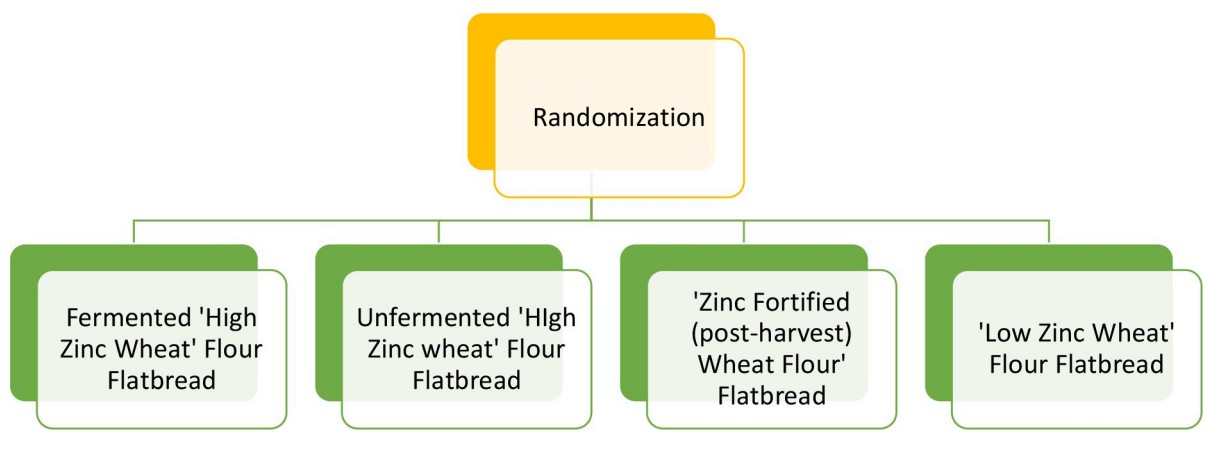

**Fig 2. Study groups.**

- Group 2: will receive unfermented high zinc agronomically biofortified wheat (>35mg/kg) flour flatbread.

- Group 3: will receive fortified (post-harvest) whole wheat (80mg/kg) flour flatbread.

- Group 4: will receive low zinc conventional whole (20–25 mg/kg) wheat flour flatbread.

The intervention duration will be six months, and participants will be provided with the respective 'flatbread' with a vegetable curry or pulse (*daal*) meal once a day for six days a week for six months This meal would be distributed at schools/colleges or community spaces during break or after the school is over.

## Outcomes

**Primary.**

- Serum zinc concentration

- Zinc deficiency

- HbA1C

- HOMA-IR (cut-off >2) [24]

- Lipid profile (total cholesterol, very low-density lipoprotein (VLDL), low-density lipoprotein (LDL), high-density lipoprotein (HDL); triglycerides (TGs))

**Secondary.**

- Body Mass Index (BMI)—Severe thinness, thinness, normal weight, overweight, obese

- Red blood cell membrane fatty acids concentrations

- Anemia

- Blood Pressure

- School attendance

- Morbidity–Diarrhea, Acute respiratory infection (ARI) etc.

**Compliance.**

- Number of days each participant had flatbread.

- Flatbread consumed (weight) each day.

## Eligibility criteria

The study will include adolescents (aged 10–19 both male and female) and adult women (aged 20–40). Participants would be eligible if they can consume wheat flour (no celiac disease), are not severely malnourished, pregnant, or lactating at enrollment, with no known chronic disease, and not enrolled in any other nutrition program or taking micronutrient supplements. Individuals planning to leave the study site during the study period will also not be eligible. The enrollment has started in Aug 2023.

For the adult women, an additional eligibility criterion will be to assess their risk for T2DM. These risk factors were modified from an existing risk assessment scale, FINDRISC (Finnish Diabetes Risk Score) and Type 2 Diabetes Risk Test by American Diabetes Association (ADA) [25, 26]. Adult women with any one of the following risk factors will be eligible to be enrolled in the study.

- BMI ($>25kg/m^2$)

- Waist circumference ($> 31$ inches)

- Family history of diabetes

- HbA1c ($>5.1\%$)

- FBG ($>100mg/dl$)

- Homeostatic Model Assessment of Insulin Resistance (HOMA-IR) ($>2$)

The HbA1c threshold for prediabetes are lower in Pakistan population compared with international guidelines, hence we have kept this as 5.1% [27].

## Study site

The study will be conducted in Mithi located in district Tharparkar in the province of Sindh, Pakistan. Tharparkar is an impoverished district of Sindh, characterized by a tropical desert climate and is located about 320 kms east from the provincial capital Karachi [28], and has a total area of 19,637 $km^2$ and a population of 1,647,036 (rural: 1,514,502; urban: 132,534) of which 53.4% are males and 46.4% are females [29]. A staggering 87% of its population lives below the poverty line [30], due to limited economic opportunities and dependence on seasonal rains [31]. Administratively, it is divided in seven talukas and subdivided in 44 union councils [28]. The district experiences erratic annual rainfall, sometimes as low as 100mm, leading to droughts [32]. In Sindh about 58.8% of the adolescent girls are anemic of which 63.1% are from rural districts of Sindh, while 21.4% women aged 15–49 years old have zinc deficiency [1]. Tharparkar has high food insecurity [1], with 60% of the children under five stunted, 33.3% wasted, 19.8% with both stunting and wasting and 40.4% of the non-pregnant women of reproductive age are underweight [1]. The population of Mithi are predominantly vegetarians.

We have selected two high schools in Mithi where we would enroll adolescents and an elementary college, midwifery college and the neighboring community for adults 20–40 years of

age. A list of current students from the respective school/colleges and individuals residing in the community (not attending school) will be collected through household line listing. All individuals on this list will undergo an eligibility assessment and will be individually randomized to the four groups.

We will obtain written informed consent from all eligible participants, and assent from participants under 18 years of age and consent from their caretakers. Upon receiving consent, the list of eligible adolescents and female adults will be randomly assigned to one of four groups, which would be blinded, and color coded as red, blue, green, and black.

## Sample size and sampling strategy

**Sample size calculation.** The study sample size was calculated based on an individually randomized four arm design on the primary outcomes of mean serum zinc levels and HbA1c. The mean HbA1c taken was 5.62% (SD 1.96) [33] and mean serum zinc was 79.5 μg/dL (SD 35.9) according to NNS-2018 [1]. The sample size was determined with a power of 0.8 and alpha of 0.05 to detect a difference of at least 0.12 effect size in the mean serum zinc levels between groups and to detect a 0.2 effect size in the mean HbA1c levels with a dropout rate of 10%. The sample size calculated was 250 participants in each group, with 210 adolescents (aged 10–18 years) and 40 adult women (aged 20–40 years). Hence, a total of 1000 participants would be randomly allocated to each of the groups in a 1:1:1:1 ratio.

**Randomization and allocation concealment.** Individuals will be randomly assigned in the four groups (red, blue, green, and black) through a computer-generated random number list. A statistician will assign the individuals into four groups by a computer-generated list and would then assign color codes (red, blue, green, black) without knowing which color belongs to which group. This randomization will be stratified on gender and age (10–13 years, 14-19years, 20–40 years old) and adjusted for serum zinc, BMI, zinc deficiency and HbA1C.

An independent consultant from an international University (University of Sydney) would be responsible for assigning the three intervention and control groups to each color group and will not be involved in the trial, including recruitment, implementation, or evaluation. The consultant would be responsible for generating and safeguarding the code and would be the only person communicating with the flour mill (Shoaib Corporation) and nutrition laboratory to ensure that the correct flour is being milled and packed in the right color bag for each batch. The consultant will break the code when the final data is collected and analyzed.

This flour mill site (Saleh Pat) is located about 500km from the trial site and the flour would be transported fortnightly to the field site. Shoaib Corporation has set up a separate local mill (chakki) for the study and has the warehouse where the two different varieties of wheat are being stored. They would be responsible to mill, fortify (ZnO - 80mg/kg) and pack high zinc, fortified, and low zinc wheat/flour in 40kgs polypropylene sacks, corresponding to a specific color. To ensure accuracy, fortnightly samples of color-coded products will be sent to the laboratory and the results will be shared with the consultant for confirmation.

**Blinding.** Participants, study personnel, and the outcome assessors will be kept unaware of the study groups. All aspects of the study, from randomization to implementation, data collection, and analysis, will be conducted using color codes assigned by an independent statistician. Color codes will be applied to wheat bags, ensuring packing, storage, and transportation in appropriately colored containers. Once participants are randomized, they will receive a laminated ID card with their photo, with the card's color indicating their assigned group to ensure that participants do not attempt to swap groups. A study kitchen is set up at the field site and has separate color-coded tandoors (ovens) for each group and would also have separate teams preparing these flatbreads who will wear color-coded aprons. Lunchboxes containing flatbread

and curry will also be color-coded and packed at the study kitchen and transported to respective schools/colleges and designated community spaces.

## Wheat cultivation and product development

**Wheat cultivation.** The cultivation of wheat to be used for this study was carried out at Saleh Pat by Shoaib Corporation and appropriate land was selected. Two cultivars, namely '*Faisalabad-2008*' *(*agronomically biofortified*)* and '*TD-1*' (conventional) were selected by consultation with local experts. After harvesting, the wheat was homogenized for both the variants and stored in the warehouse under standard conditions. Agronomic biofortified was produced by zinc foliar application and was sprayed two times with $ZnSO_4 \cdot 7H_2O$ at 15 days interval at the heading stage. The conventional low wheat variety was grown in the same conditions, but without spraying.

**Product development.** The post-harvest fortified wheat flour was prepared by adding 80 ppm of zinc to 1kg of conventional 'low zinc' variety. The fortification was performed in small batches every 15 days. Fortification was carried out in two steps, first by making a concentrated premix. Initially 3.2 g of Zinc oxide was mixed with 1kg of flour and subsequently added this premix to 19kg conventional wheat in a rotation mixer and mixed and then further 20kg of wheat flour was added to make 40kg bag.

Product development and sensory testing were conducted at the Bahauddin Zakariya University in Multan and the Mithi field site. Each product was developed at different extraction rates and variable fermentation times, considering local contextual factors. The whole wheat product of both varieties and two-hours fermented flatbread were deemed acceptable by the panelists during sensory testing and hence were chosen for the trial. The details would be published in a separate paper.

## Intervention delivery

**Flatbread preparation, and delivery.** The flatbread and the vegetable/pulses curry will be prepared in the study kitchen established at the study site and professional catering staff will be hired to prepare food following standard guidelines. Four color-coded tandoor ovens will be set up, and the designated kitchen staff for each group will wear color-matched uniforms. The regular menu will feature flatbreads with a choice of vegetable curry or pulses, designed weekly and prepared in the study kitchen to ensure consistency.

High zinc fortified flour and low zinc wheat flour will be mixed separately with water, salt, oil in their respective color-coded utensils. The ingredients will be kneaded by hand to form soft dough. The dough will then be covered with a muslin cloth and the resting time of the dough will be kept identical for each of the produced flour portions (30 min) (Table 1) For

**Table 1. Ingredients for flatbread.**

| Ingredients | Unit | Quantity | |
|---|---|---|---|
| | | Fermented | Unfermented |
| Wheat flour | g | 1000 | 1000 |
| Active dry yeast | g | 3.50 | - |
| Iodized Salt | g | 4 | 4 |
| Ghee/cooking oil | g | 30 | 30 |
| Water for kneading the dough | ml | 300 | 300 |
| One ball of dough | g (±5g) | 125.00 | |
| Weight of one ready cooked flatbread | g (±5g) | 100.00 | |

fermented flour, we will add 3.5g yeast /kg of flour and let the dough rest for 2hrs at 30 degrees Celsius.

Each ball will be weighed on a weighing machine to ensure that each ball has a weight of 120 grams. It will be transferred to a baking tray or round cushion lined and these will be used to place the flatbreads inside separate tandoors for each group. To allow for proper rising, flatbreads will be spaced adequately inside the tandoor. Prepared flatbreads will be packed in color coded lunch boxes and each box will have two flatbreads (each flatbread divided in four equal parts) and curry under supervision. These boxes will then be transported to the targeted schools/colleges/ community space and packing will ensure that the freshness of flatbread is maintained.

**Procedure of serving flatbread/curry in schools.** Field staff will deliver the respective lunchbox to the specific group in each school. The adolescents in different groups in a single school/college would be segregated and seated in separate marked rooms. The lunch boxes would be delivered after matching participants' specific IDs on the box with the student badges. The staff will also have extra flatbreads in color-coded bags and would give to individuals who would demand for more. The study team will record the amount of flatbread consumed by each participant for each day and data recorded electronically on an android app. Research staff and teachers will ensure each participant receives their designated lunch, record food consumption, and maintain order, ensuring that food is not being shared or taken home.

## Data collection

Baseline and endline measurements will be conducted for all participants, whereas for midpoint assessments, sampling scheme will be assessed by randomized, arm-based stratified selection so that a third from each arm is tested at two additional time points.

For all study visits, electronic app-based questionnaires will be used to guide personnel in conducting structured interviews with the study participants at baseline and endline. Data on 24-hr dietary recall will be collected in paper-based questionnaires. A summary of all monitoring and visit-specific forms will be used throughout the study.

## Hiring and training

All staff would be hired after a rigorous scrutiny process and a preference would be given to local people as they are most aware of the local customs and language. Upon recruiting, the research staff would undergo a comprehensive eight-day training and refresher training at midline and before endline assessments. The training will cover areas of line listing, mobile app data collection, anthropometric measurements, blood pressure measurements, blood sampling, food handling and ethical conduct and adherence to regulations. Kitchen staff will receive comprehensive training on handling wheat, flour, dough, flatbread preparation, weighing methods, hygiene, and food packing.

## Pilot testing

Survey questionnaires will be pilot tested with a small group of participants (maximum 50) for response latency, question interpretation and appropriateness. Face validity and construct validity for all questionnaires will also be conducted with a small expert group with knowledge in public health and nutrition epidemiology.

## Baseline, midline, and endline data collection

Participants will undergo assessments at baseline and endline, where we will gather data on socio-demographic factors, dietary intake via 24-hour dietary recall, anthropometric

measurements (weight, height, MUAC, waist circumference), blood pressure measurements, and blood samples will be collected for biochemical analysis, including serum zinc levels, FBS, HbA1C, and insulin at both time points. Lipid profile and RBC membrane fatty acids will be assessed only at the endline. At midline, we will conduct 24-hour dietary recalls, measure serum zinc and HbA1C for a subset from each group. The data for morbidity (diarrhea and ARI) will be collected fortnightly for the entire duration of the intervention.

**Demographic and socioeconomic indicators of households.** Information on socio-economic status, gender, ethnicity, level of education, marital status, and occupation of the household head, water, sanitation, hygiene (WASH) and food insecurity will be collected. Household information will be captured from the head of the household or any knowledgeable member of the household (aged 18 years or more).

**Dietary intake.** Dietary intake will be assessed using the USAID FANTA Household Dietary Diversity Scale (HDDS) and a 24-hour dietary recall questionnaire. The 24-hour recall will be conducted for the entire sample at baseline, midline, and endline, covering weekdays and weekends to capture dietary variability. Detailed food information will include description, additives, combinations, brand (if relevant), quantity, time, occasion, source, and consumption location (home or elsewhere). Interviewers and staff will be familiar with local foods, aided by guidance documents. Local utensils and portion-size photographs will assist in estimation.

**Morbidity.** Incidence and duration of diarrheal episodes and respiratory tract infections will be recorded every two weeks throughout the study.

**Anthropometric measurements.** Trained research staff will conduct anthropometric assessments at households and schools. Weight and height measurements will be recorded to the nearest 0.1 kilograms and centimeters, respectively. This will be done with participants in light clothing and without shoes, using a Seca digital floor scale (model 813) and Seca stadiometer (model 213). Mid-Upper Arm Circumference (MUAC) and waist circumference will be determined using standardized procedures and a MUAC measuring tape (Seca 201). All measurements will be taken in duplicate by two research staff members, and if discrepancies exceed 1 cm for height, 0.5 kg for weight, or 0.5 cm for MUAC/waist circumference, a third measure will be taken by the team leader and recorded using standardized procedures.

**Biochemical sample collection.** A certified phlebotomist will supervise blood collection. 23-mm gauge needles will be used for adolescents and women of reproductive age, with trace element-free tubes for blood collection. Used needles, sharps, and other consumables will be disposed-off safely. Strict personal hygiene will be maintained, including the use of disposable gloves and hand sanitizer. Skin cleaning and vein visibility will be ensured, and 5 ml of venous blood will be collected. Samples will be labelled with barcoded stickers. After labeling, the collection tubes will be left undisturbed for 30 minutes, and then centrifuged at 3000 rpm for 10 minutes. Serum (at least 1.0 ml) will be transferred to pre-labelled tubes and then placed in zip-lock bags. These serum samples will be stored in cool boxes at 2 to 8˚C with the inclusion of ice packs. Used consumables will be safely disposed of.

## Data analysis

**Biochemical variables.** Analysis of samples will be conducted at Nutrition research lab, Aga Khan University, where external quality assurance (VITAL-EQA) is managed by Centers for Disease Control and Prevention (CDC). Individual biomarker concentrations will be used to determine deficiency using standard cut-offs for age and sex. CRP concentrations will be used to identify inflammation for all participants. We will check whether adjustment is necessary for serum zinc concentration and will be adjusted according to the BRINDA project

guidelines to determine zinc deficiency (VAD) and if so present both adjusted and unadjusted values. Fasting blood glucose and blood Hb1C levels will be measured using latex agglutination inhibition and photometric assays, respectively on cobas c 311 biochemistry analyzer (Roche Diagnostics). Serum zinc levels will be measured using flame atomic absorption spectroscopy on Thermo Fisher iCE 3300 AAS Atomic Absorption Spectrometer.

**Food testing.** Zinc is quantified in wheat seeds using Flame Atomic Absorption Spectrometry. Reagents include a zinc master standard (1000 mg/L), de-ionized water, nitric acid, hydrochloric acid, plasma zinc controls, and wheat flour SRM NIST 1567B. The sample preparation involves grinding the seeds into a fine powder, wet digestion with an acid mixture, and filtration. The assay calibration is performed using a linear graph method with working standards (0.03 ppm to 0.50 ppm) and a blank solution. For zinc analysis, an (air/acetylene) flame at a flow rate of 1.0 L/min, providing a temperature of 2500°C, is utilized for the atomization of gaseous zinc molecules. The zinc analysis further involves the detection wavelength of 213.9 nm, utilizing an automatic monochromator calibration with both the deuterium lamp and the zinc hollow cathode lamp. The analysis of filtered wheat seed samples is performed without further dilution, with each sample run in triplicates. Blank solutions are prepared, and standard deviation calculations ensure data reliability. Plasma controls and NIST wheat flour SRM (1567B) are included in each batch to ensure accuracy.

The quantification of phytate content in wheat seeds holds significance for gaining insights into nutritional profiles and refining food processing techniques. To accomplish this, the Megazyme Kit (K-PHYT 05/19) was utilized, providing a methodical framework that incorporates essential reagents like buffers, phytase and alkaline phosphatase suspensions, and phosphorus standards. The preparation of the phosphorus calibration curve involves creating standard phosphorus solutions with specified volumes and DI water. The subsequent calculation process determines the absorbance (A655) for each standard, leading to the derivation of ΔA phosphorus and mean M, which is then used to calculate the phosphorus content of test samples. Sample extraction is started by adding 0.5g of the ground wheat sample into a beaker and then10.0mL of 0.66M HCl is pipetted into the beaker. This mixture is then stirred vigorously using an electronic magnetic bead stirrer for at least 3 hours at room temperature. After stirring the mixture for a minimum of 3 hours, the clear extract obtained by centrifugation at 13000 RPM for 10 minutes is neutralized using 0.75M NaOH (1:1) for the enzymatic dephosphorylation reaction. The colorimetric determination of phosphorus involves pipetting 1.00 mL of the sample, control, or phosphorus standard, adding 0.50 mL of the color reagent, and incubating in a water bath at 40°C for 1 hour. The absorbance at 655 nm (A655) is then read within 3 hours. For phosphorus/phytic acid determination, absorbance values for "Free Phosphorus" and "Total Phosphorus" samples are obtained, and their concentration is calculated using a formula considering mean M, dilution factor, absorbance change, and relevant factors. The entire process ensures accurate and reliable results for phosphorus and phytic acid analysis in wheat seeds.

**Anthropometric variables.** Anthropometric variables including the average height (m), weight (kg), waist circumference and MUAC would be collected in duplicate and the average (mean) of acceptable paired measures will be used in the analysis. Participant Body Mass Index (BMI) will be calculated and converted to BMI-for-age z-scores (BAZ) along with height-for-age z-scores (HAZ) according to chronological age using the WHO Growth Reference for Adolescents and WHO-package for R. Participants with a HAZ <-2 SD will be classified as stunted; severe thinness with a BAZ $\leq$ -3 SD; thinness with a BAZ > -3 SD to $\leq$-2 SD, normal weight with a BAZ >-2 SD to <+1SD, overweight with a BAZ >+1 SD to < +2 SD and, obese with a BAZ $\geq$+2 SD. The prevalence of anthropometric indicator categories will be reported as the proportion of participants who did not achieve the respective cut-off. MUAC

will be assessed using cut-offs outlined in the Integrated Management of Adolescent and Adult Illness (WHO) as the proportion of participants with MUAC < 160 mm.

**Food security and 24-hour dietary recall.**   Using the eight dichotomous questions (yes/ no) within the Household FIES, households will be classified based on the total number of affirmative responses ranging from 0–8. As recommended by FAO, Rasch modelling techniques will be used.

The data collected from 24-hour recall will reveal the nutrient consumption patterns of each child beyond the study-provided food, aiding in assessing the actual intervention impact. Given the lack of a food composition database for Pakistan, we intend on using a database compiled by a recent study in Pakistan (MAL-ED) which compiled nutrient data from various sources. This comprehensive database includes foods from multiple references, including World Food Dietary Assessment System, USDA National Nutrient Database, NUTTAB online database, Composition of Foods Integrated Database, and the Food Composition Table for Bangladesh. If values could not be located through these sources, values were calculated using common recipes by either the PKN or INV sites of the MAL-ED study [34–44]. The final contents of the MAL-ED Pakistan Food Composition Table consist of a combination of individual food items pulled from the sources referenced above, and nutrient values derived from calculating nutrients in the recipes collected as part of the study.

A trained nutritionist will assign food codes, and retention factors will be applied to account for nutrient changes due to different cooking methods. The mean total intake of each nutrient will be calculated using frequency weight and nutrient content for each food.

## Trial analysis

This trial will adhere to CONSORT guidelines for randomized trials, with primary analysis on an intention-to-treat basis [22]. Both primary and secondary data will be analyzed following the same principle. Descriptive statistics will include mean and standard deviation for symmetrically distributed continuous variables, median and interquartile range for asymmetrically distributed ones, and frequency with percentages for categorical data. Associations between dependent and independent categorical variables will be accessed using chi-square test, while one-ANOVA will be used to assess difference in continuous outcomes by intervention groups. Significant association further assessed by post-hoc test adjusting for multiple comparisons. Multivariate analysis will be performed using mixed effect linear or logistic regression to assess the impact of intervention over time by including time and group interaction. The time and intervention group will be treated as fixed effect while students included in the model as random intercept. The potential confounding factors that are imbalance at baseline will also be included as fixed effect in the model. The minimal adequate model for regression will be computed using the backward procedure. Statistical Significance will be set at p-value of <0.05. To assess the robustness of the findings, a sensitivity analysis will be conducted, and a dose-response relation will also be assessed.

## Data management

Data collection will be electronic, using Android OS devices with a customized Java application, incorporating data quality checks in real-time. Data will be transferred daily to the Aga Khan University (AKU) server via the internet or manually via USB in areas with no internet access. Password protection will restrict data access, with encryption and anonymization for confidentiality. An AKU repository will store data, accessible only by authorized personnel through AKU-LAN identification. A backup and fail-over server will ensure data security. Documentation including an installation guide, user manual, and database documentation

will facilitate data transfer. Field supervisors will conduct spot checks, fortnightly refresher sessions, and monitor 10% of participants. Laboratory procedures will be quality-assessed through result rechecks and standardization.

## Ethical considerations

This study will secure ethical approval from both the Aga Khan University Ethical Review Committee (ERC) and the National Bioethics Committee (NBC), Pakistan (S1 Checklist). We will consistently uphold ethical principles, ensuring autonomy, anonymity, confidentiality, and equity throughout the study. Informed consent will be obtained from all eligible participants, with parental or caregiver consent for those under 18, alongside assent. Collaborations with the Education and Health departments of the district will be closely maintained. Participants diagnosed with T2DM and other issues will be referred to appropriate health facilities. Any adverse event due to the intervention (though unlikely) will be taken care by the study. We will conduct regular community engagement events and information sessions during the intervention to address concerns and encourage continued participation. Participants will be free to withdraw from the trial at any point. Ethical conduct with trial participants, especially children, will adhere to UNICEF's ethical guidelines [45]. These principles emphasize the utmost respect for every child's dignity and rights, ensuring privacy, confidentiality, and participation in decisions that affect them. All blood samples and other reports will be shared with the participants. Any changes to the protocol will be informed to the ethical review committee.

## Discussion

Micronutrient deficiency is a serious health concern, which could cause serious health conditions. Malnutrition is a global issue as nearly a quarter of the world's population is suffering from one or more micronutrient malnutrition disorders. Globally, 22.2% children are stunted, 50.5 million are wasted, and 38.3 million under five children are overweight due to multifaceted malnutrition [46]. The issue of malnutrition is persistently high with slow progress in many developing regions of the world. Zinc deficiency is amongst the common micronutrient deficiency and a major global health challenge, as approximately 17% of the global population does not get enough zinc in their diet. Pakistan has one of the highest rates of zinc deficiency worldwide, affecting 20–40% of the population, with large provincial and regional disparities.

Amongst the many strategies to alleviate zinc deficiency, food fortification is considered a safe and cost-effective measure to prevent zinc deficiency amongst other micronutrient deficiencies. Biofortification of food crops (enhancing micronutrient content using plant breeding techniques) is also a common fortification intervention to increase provitamin A, zinc, iron, and folate contents in staple foods. Agronomic biofortification also increases the crops' micronutrient mobilization and utilization potential. It is reported to be produced and consumed by over 20 million people globally [47]. The EZZAFAW trial is an evaluation of the efficacy of zinc fortification and will help inform policy for the most suitable strategy to deliver zinc, as this offers a direct comparison of agronomically biofortified and fortified wheat and explores the additional effects of fermentation.

T2DM poses a major public health challenge, due to its increasing prevalence and complex complications. In populations with reduced growth and thin physiques, like in India, there is an elevated risk of truncal obesity and metabolic issues due to sedentary behavior and high-glycemic diets, exacerbated by low birth weight [48]. Chronic glucose exposure affects erythrocyte membranes during their long lifespan, leading to structural and functional disruptions.

These whole wheat flatbreads offer additional health benefits, making the findings applicable to advocacy and potential scale-up in this context and whole wheat was selected as the preferred choice since it aligns with the community's customary practice of grinding their wheat or purchasing it from a local chakki. In summary, whole-wheat flour that has been fortified by plant breeding or industrially is an ideal food to deliver more zinc to the general population of Pakistan. Adoption and acceptability of biofortified wheat is also expected to be high since zinc does not affect wheat's sensory characteristics, and the yield of varieties available in the Pakistani seed systems is competitive with other varieties currently grown by farmers in Pakistan. In addition, fermentation of the flour to make naan (leavened flatbread) can increase the bioavailability of zinc. Using this low-cost and culturally acceptable food processing technique, the amount of zinc absorbed from biofortified and fortified wheat can be increased significantly and sustainably—a double duty action to help improve absorbable dietary zinc intake, reduce zinc deficiency, and lower the risk of some of the most common non-communicable diseases among adolescents and young adults.

## Supporting information

**S1 Checklist. SPIRIT checklist and protocol submitted for ERC.**
(DOCX)

## Acknowledgments

We would like to acknowledge the support of the funders. We would also like to acknowledge the support of the various departments of Aga Khan University.

## Author Contributions

**Conceptualization:** Jai K. Das, Zulfiqar A. Bhutta.

**Funding acquisition:** Jai K. Das.

**Investigation:** Tariq Ismail, Saeed Akhtar, Kehkashan Begum, Junaid Iqbal.

**Methodology:** Jai K. Das, Zahra Ali Padhani, Muhammad Khan, Mushtaq Mirani, Arjumand Rizvi, Imran Ahmed Chauhadry, Tariq Ismail, Saeed Akhtar, Kehkashan Begum, Junaid Iqbal, Khadija Humayun, Zulfiqar A. Bhutta.

**Project administration:** Muhammad Khan, Mushtaq Mirani.

**Writing – original draft:** Jai K. Das, Zahra Ali Padhani.

**Writing – review & editing:** Muhammad Khan, Arjumand Rizvi, Imran Ahmed Chauhadry, Rahima Yasin, Tariq Ismail, Saeed Akhtar, Kehkashan Begum, Junaid Iqbal, Khadija Humayun, Hamna Amir Naseem, Kauser Abdulla Malik, Zulfiqar A. Bhutta.

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
