## [Decision Letter · Decision Letter 0]

26 Apr 2024

PONE-D-23-42882Efficacy of Zinc Fortified and Fermented Wheat Flour: A Randomized Controlled Trial ProtocolPLOS ONE

Dear Dr. Das,

Thank you for submitting your manuscript to PLOS ONE. After careful consideration, we feel that it has merit but does not fully meet PLOS ONE’s publication criteria as it currently stands. Therefore, we invite you to submit a revised version of the manuscript that addresses the points raised during the review process.

We look forward to receiving your revised manuscript.

Kind regards,

Fernando Henrique da Silva Reboredo, Ph.D

Academic Editor

PLOS ONE

 [International Food Policy Research Institute (IFPRI) - (HarvestPlus Program)].  

Additional Editor Comments:

Dear Authors

According the remarks of both reviewers minor revisions must be implemented in your manuscript.

Please take into account all the comments in order to proceed to final stage of the publication process.

Best regards

Reviewers' comments:

Reviewer's Responses to Questions

**Comments to the Author**

1. Does the manuscript provide a valid rationale for the proposed study, with clearly identified and justified research questions?

Reviewer #1: Yes

Reviewer #2: Yes

2. Is the protocol technically sound and planned in a manner that will lead to a meaningful outcome and allow testing the stated hypotheses?

Reviewer #1: Yes

Reviewer #2: Yes

3. Is the methodology feasible and described in sufficient detail to allow the work to be replicable?

Reviewer #1: Yes

Reviewer #2: Yes

4. Have the authors described where all data underlying the findings will be made available when the study is complete?

Reviewer #1: Yes

Reviewer #2: Yes

5. Is the manuscript presented in an intelligible fashion and written in standard English?

Reviewer #1: Yes

Reviewer #2: Yes

6. Review Comments to the Author

You may also provide optional suggestions and comments to authors that they might find helpful in planning their study.

Reviewer #1: Thank you for the opportunity to review this paper. This is an interesting manuscript presenting a protocol of a randomized controlled tral on effecacy of Zinc Fortified and fermented wheat flour. My review mainly concerns only the statistical aspects of the study. Some questions reported below were raised and in my view, it is not acceptable in this form for the publication in this journal.The "trial analysis" paragraph should be more detailed. Particularly in relation to multivariate models. Authors should anticipate Intention-to-treat analysis of the missing data and possible sensitivity analyses.

Reviewer #2: Upon reviewing the manuscript titled "Efficacy of Zinc Fortified and Fermented Wheat Flour (EZAFFAW): A Randomized Controlled Trial Protocol" (Manuscript Number: PONE-D-23-42882), it became evident that the authors have outlined a comprehensive and timely protocol that addresses the urgent need of developing strategies to mitigate zinc deficiency among the population, particularly adolescent and young women living in economically disadvantages areas. The proposed protocol includes 1000 participants, allocated by stratified randomization to four groups. Zn supplementation of the daily diet via its incorporation into bread, a common food for the population appears to be a promising option.

Monitoring the impacts on morbidity, anthropometric and biochemical indicators, and other key metrics is ensured through the evaluation of multiple indicators.

The processes for obtaining flour that has been fortified with ZnO or biofortified via foliar spraying with ZnSO4.7H2O is comprehensively elucidated, along with the process of bread preparation. Nonetheless, I would like to pose a few questions:

1. Why was not the same variety of wheat used for the production of both agronomically biofortified and biofortified flour?

2. During the preparation of the bread, the dough was left to rest for 30 minutes or more (depending on whether it was fermented or not) at what temperature?

7. In the section "Data analysis - Food testing" the quantification of phytate includes a stirring period of 3 hours. Regarding the technique and temperature involved, I propose that it be clarified more deeply.

7. PLOS authors have the option to publish the peer review history of their article (what does this mean?). If published, this will include your full peer review and any attached files.

Reviewer #1: No

Reviewer #2: No

---

## [Author Response · Author response to Decision Letter 0]

2 May 2024

Reviewer #1: Thank you for the opportunity to review this paper. This is an interesting manuscript presenting a protocol of a randomized controlled trial on efficacy of Zinc Fortified and fermented wheat flour. My review mainly concerns only the statistical aspects of the study. Some questions reported below were raised and in my view, it is not acceptable in this form for the publication in this journal. The "trial analysis" paragraph should be more detailed. Particularly in relation to multivariate models. Authors should anticipate Intention-to-treat analysis of the missing data and possible sensitivity analyses.

Response: we would like to thank the reviewers for their review and encouragement and as suggested, we have added the required details in the data analysis section.

Reviewer #2: Upon reviewing the manuscript titled "Efficacy of Zinc Fortified and Fermented Wheat Flour (EZAFFAW): A Randomized Controlled Trial Protocol" (Manuscript Number: PONE-D-23-42882), it became evident that the authors have outlined a comprehensive and timely protocol that addresses the urgent need of developing strategies to mitigate zinc deficiency among the population, particularly adolescent and young women living in economically disadvantages areas. The proposed protocol includes 1000 participants, allocated by stratified randomization to four groups. Zn supplementation of the daily diet via its incorporation into bread, a common food for the population appears to be a promising option.

Monitoring the impacts on morbidity, anthropometric and biochemical indicators, and other key metrics is ensured through the evaluation of multiple indicators.

The processes for obtaining flour that has been fortified with ZnO or biofortified via foliar spraying with ZnSO4.7H2O is comprehensively elucidated, along with the process of bread preparation. Nonetheless, I would like to pose a few questions:

1. Why was not the same variety of wheat used for the production of both agronomically biofortified and biofortified flour?

Response: we would like to thank the reviewers for the thorough review and comments. We liaised with various experts and as Akber-19 has been introduced as a zinc enriched variety and locally developed, so we wanted to compare this against a conventional variety and hence chose to use two different varieties.

2. During the preparation of the bread, the dough was left to rest for 30 minutes or more (depending on whether it was fermented or not) at what temperature?

Response: thanks, and we have now added this.

3. In the section "Data analysis - Food testing" the quantification of phytate includes a stirring period of 3 hours. Regarding the technique and temperature involved, I propose that it be clarified more deeply.

Response: we have now added the details here.

---

## [Decision Letter · Decision Letter 1]

14 May 2024

Efficacy of Zinc Fortified and Fermented Wheat Flour: A Randomized Controlled Trial Protocol

PONE-D-23-42882R1

Dear Dr. Das,

We’re pleased to inform you that your manuscript has been judged scientifically suitable for publication and will be formally accepted for publication once it meets all outstanding technical requirements.

Kind regards,

Fernando Henrique da Silva Reboredo, Ph.D

Academic Editor

PLOS ONE

Additional Editor Comments (optional):

Reviewers' comments:

Reviewer's Responses to Questions

**Comments to the Author**

1. Does the manuscript provide a valid rationale for the proposed study, with clearly identified and justified research questions?

Reviewer #2: Yes

2. Is the protocol technically sound and planned in a manner that will lead to a meaningful outcome and allow testing the stated hypotheses?

Reviewer #2: Yes

3. Is the methodology feasible and described in sufficient detail to allow the work to be replicable?

Reviewer #2: Yes

4. Have the authors described where all data underlying the findings will be made available when the study is complete?

Reviewer #2: Yes

5. Is the manuscript presented in an intelligible fashion and written in standard English?

Reviewer #2: Yes

6. Review Comments to the Author

You may also provide optional suggestions and comments to authors that they might find helpful in planning their study.

Reviewer #2: Thank you to the authors for including the recommended suggestions.

After receiving clear explanations for my questions, I believe that the paper is now suitable for publication.

7. PLOS authors have the option to publish the peer review history of their article (what does this mean?). If published, this will include your full peer review and any attached files.

Reviewer #2: No

---

## [Editor Report · Acceptance letter]

21 May 2024

PONE-D-23-42882R1 

PLOS ONE

Dear Dr. Das, 

I'm pleased to inform you that your manuscript has been deemed suitable for publication in PLOS ONE. Congratulations! Your manuscript is now being handed over to our production team.

Kind regards, 

on behalf of

Professor Fernando Henrique da Silva Reboredo 

Academic Editor

PLOS ONE